# IS21 family transposase cleaved donor complex traps two right-handed superhelical crossings

Mercedes Spínola-Amilibia [1], Lidia Araújo-Bazán [1], Álvaro de la Gándara[1], James M. Berger [2] & Ernesto Arias-Palomo [1] ✉

Transposases are ubiquitous enzymes that catalyze DNA rearrangement events with broad impacts on gene expression, genome evolution, and the spread of drug-resistance in bacteria. Here, we use biochemical and structural approaches to define the molecular determinants by which IstA, a transposase present in the widespread IS21 family of mobile elements, catalyzes efficient DNA transposition. Solution studies show that IstA engages the transposon terminal sequences to form a high-molecular weight complex and promote DNA integration. A 3.4 Å resolution structure of the transposase bound to transposon ends corroborates our biochemical findings and reveals that IstA self-assembles into a highly intertwined tetramer that synapses two super-coiled terminal inverted repeats. The three-dimensional organization of the IstA•DNA cleaved donor complex reveals remarkable similarities with retro-viral integrases and classic transposase systems, such as Tn7 and bacteriophage Mu, and provides insights into IS21 transposition.

The transposition of mobile DNA elements – nucleic acid sequences that can move from one locus to another in the genome – is a strong evolutionary force and a source of genetic instability and diversity[1,2]. Transposases, one of the most abundant genes in nature[3], are the catalyst of this fundamental process. The DNA reorganizing action of transposases can modify gene expression, spread antibiotic resistance genes and toxins, and has been linked to numerous human diseases[4–9]. Transposases also share a common ancestor with central enzymes of the retroviral integration process and the CRISPR-mediated immune response[10,11]. Some transposases have even been domesticated to give rise to new cellular functions, such as V(D)J recombination, while other are showing potential in biotechnological and gene editing applications[12–15].

How transposition is controlled at molecular level has remained a poorly understood question. Transposition frequently occurs through a series of carefully orchestrated DNA strand cleavage and integration reactions to prevent chromosome breaks and lethal genome rearrangements. Transposases assemble, sometimes in conjunction with specific regulatory factors, into dynamic complexes

that undergo multiple conformational changes to modulate DNA break formation and ensure that, once started, the transposition reaction goes to completion (Fig. 1a)[16,17]. These enzymes are typically tasked with: (1) selectively recognizing a set of terminal, inverted DNA sequence repeats (TIRs) present at both transposon ends, (2) synapsing the TIRs into a 'paired-end' complex, (3) cleaving one or both strands of each transposon end, (4) capturing a target DNA, and (5) performing a strand transfer reaction to insert the mobile element into the new site. Strikingly, although most transposases share a similar function, their precise catalytic pathway and three-dimensional architectures can be remarkably divergent[18,19].

Numerous transposases contain a 'DDE'-family metal-binding domain, which can also be found in closely related retroviral integrases[10,20]. The simplest and most abundant autonomous DDE transposons are referred to as insertion sequences (ISs)[21]. IS elements are usually formed by one or two genes delimited by two TIRs. Interestingly, many ISs contain promoter sequences that can modify the expression of neighboring genes[22–24], and several families have been

[1]Department of Structural & Chemical Biology, Centro de Investigaciones Biológicas Margarita Salas, CSIC, Madrid 28040, Spain. [2]Department of Biophysics and Biophysical Chemistry, Johns Hopkins University School of Medicine, Baltimore, MD 21205, USA. ✉e-mail: earias@cib.csic.es

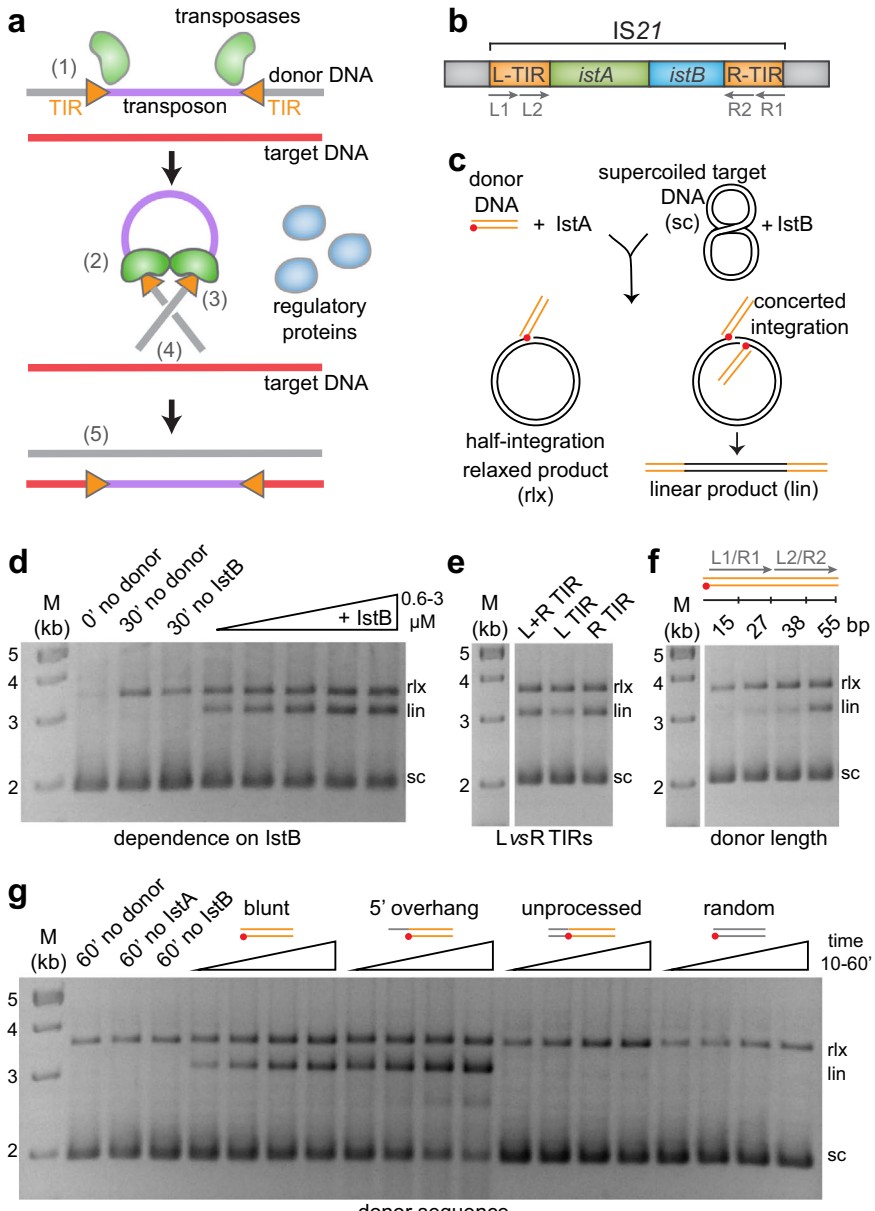

**Fig. 1 | IS21 transposon organization and characterization of the DNA integration reaction. a** Simplified schematic of the transposition reaction. TIR terminal inverted repeats. Numbers indicate some of the steps of the transposition reaction described in the main text. **b** Genetic organization of prototypical IS21 members, including IS5376. The left and right TIRs (L-TIR and R-TIR) are composed of two multiple repeats (L1/L2 and R1/R2). **c** Schematic of an in vitro integration assay. Pre-cleaved donor DNA shown in orange and target plasmid in black. Red dot indicates the position of the terminal CA dinucleotide that contains the donor 3'-OH. **d** Integration of the transposon ends performed in the absence and presence of variable IstB concentrations (equimolar mixture of pre-cleaved L an R ends). Molecular weight marker (in kb) indicated as M. Nucleic acid products labeled throughout the figures as supercoiled (sc), linear (lin) and relaxed (rlx) DNAs. This experiment was repeated three independent times giving similar results. Source data are provided as a Source Data file. **e** Integration assay with different combinations of pre-cleaved left (L) and right (R) TIRs. Molecular weight marker (in kb) indicated as M. This experiment was repeated three independent times. Source data are provided as a Source Data file. **f** Integration assay performed with four lengths of the donor DNA (mixture of pre-cleaved L an R ends). Molecular weight marker (in kb) indicated as M. This experiment was repeated three independent times. Source data are provided as a Source Data file. **g** in vitro integration reaction of multiple donor DNA sequences. TIR sequence represented in orange. Flanking or random sequences shown in grey. Red dots indicate the position of the CA dinucleotide. Molecular weight marker (in kb) indicated as M. This experiment was repeated three independent times. Source data are provided as a Source Data file.

associated with the dissemination of antibiotic resistance and the spread of multidrug-resistance[9,25,26].

The IS21 family of insertion sequences is a particularly widespread class of transposable elements. IS21 genes have been found in clinical and multidrug-resistant strains of *Escherichia coli* and *Staphylococcus aureus* and have played a significant role in the evolution of the human pathogens *Yersinia pestis* and *Y. pseudotuberculosis*[26–33]. Members of the IS21 family are composed of two open reading frames (coding for

IstA and IstB) flanked by two TIRs[34,35] (Fig. 1b and Supplementary Fig. 1a). IstA is a DDE transposase that is able to act on single and tandem IS21 elements with comparable efficiency[36,37] (Supplementary Fig. 1b). By comparison, IstB is a regulatory factor and member of the AAA + (ATPases associated with various cellular activities) family of ATPases; this protein is similar to MuB and TnsC from the Mu and Tn7 mobile elements and is essential to capture the target DNA and recruit the transposase to promote efficient strand transfer[36–43]. Although it

has been shown that IstB can self-assemble into decamers that grasp and bend target DNA segments[43], how its cognate IstA transposase recognizes the transposon ends to prevent deleterious double strand breaks and stimulate DNA transposition has been unclear. The precise architecture of the synaptic complex between IstA and transposon DNA similarly has been unresolved.

To better understand IstA and DDE family transposase mechanism in general, in this work we have used a combination of functional, biophysical, and structural methods to investigate the interaction between the IS21 transposase and its transposon ends. Using an in vitro integration assay, we show that both IstA and IstB are essential for the integration reaction. These analyses, together with biochemical studies, reveal several molecular determinants that must be present in donor DNA segments for transposition to occur, and show that IstA oligomerizes and associates with its cognate nucleic acid elements in a highly cooperative manner. Cryo-EM studies support our biochemical observations and further establish that the transposase engages the transposon ends using a mechanism akin to that of retroviral integrases. The structure additionally reveals that the IS21 transposase forms a close-knit tetramer that wraps into two plectonemically intertwined duplexes, adopting a configuration reminiscent to a Tn7-like transposome and to a transposition intermediate state described for the bacteriophage Mu system. Overall, our findings highlight how different elements of the DDE transposase specifically mediate the recognition and synapsis of donor DNA duplexes to promote transposition.

## Results

### Molecular determinants of the donor DNA

Although previous genetic and biochemical reports have analyzed some of the molecular features of IS21 transposition, the absence of an in vitro reconstitution assay using purified components has limited a more complete functional and structural understanding of this mobile element family. To address this shortfall and set the stage for subsequent structural studies, we set out to define substrate and reaction conditions that would support characteristic strand transfer reactions, such as the integration of the donor DNA into a supercoiled circular substrate. After screening several candidates, IS5376, a prototypical member of the IS21 family found in *Geobacillus stearothermophilus* that is highly similar to other members present in *Escherichia coli*, *Pseudomonas aeruginosa* and *Yersinia pestis*[34,44], was chosen for this work. Briefly, IstA was initially incubated with short linear fragments of donor DNA containing the left and right transposon ends (55 bp pre-cleaved complete TIR sequences unless indicated otherwise), which were then mixed with a supercoiled target plasmid in the presence or absence of IstB and nucleotide (Fig. 1c) (a detailed table with the DNA molecules used in this study can be found in Supplementary Table 1). Integration of one TIR results in a relaxed plasmid whereas concerted integration of two TIRs, product of efficient DNA transposition, yields a linearized plasmid.

In agreement with previous reports[36,37,42], the biochemical data show that IstA alone is insufficient to perform the strand transfer reaction, while also establishing that IstB is indeed essential for IS21 integration (Fig. 1d). To better understand the products of the insertion reaction we sequenced ten transposition events. This analysis revealed that, although some insertions appeared to be clustered close to promoter regions, IS21 (like most transposons[45]) does not show clear or strong target sequence specificity (Supplementary Fig. 1c). Consistently with previous observations[44], all insertions were delimited by 5 bp-long direct repeats, defined by the separation between the two staggered cuts that the transposase is known to generate in the target DNA during the strand transfer reaction[44]. Interestingly, sequence analysis revealed that some of the plasmid products of the reaction contained a left and a right transposon end, while others contained only two left or only two right ends, indicating that under

these in vitro conditions, where IstA is incubated with an equimolar mixture of pre-cleaved left and right short linear TIR fragments, the transposase does not appear to have a strong preference for either of the transposon ends. To test this idea, we examined the activity of IstA in reactions that contained only one or both transposon ends (Fig. 1e). The results showed that IstA is active using either a mixture of TIRs or just the isolated left or right TIR. More precisely, the activity with the left end was mildly reduced (~2-fold), whereas the activity using the right end was equivalent to that seen for the mixture of both TIRs (it has been shown that other transposases can use the isolated left and right ends with varying degrees of efficacy[46–48]).

Each transposon end, such as those found in Mu or Tn7, often contains several copies of the conserved sequence that is specifically recognized by the transposase. It has been reported that the IS21 TIRs are also composed of multiple (usually at least two) direct repeats termed L1-L2 and R1-R2 respectively[34,35,44] (Fig. 1b and Supplementary Fig. 1a). However, the exact function of these repeated sequences has not been assessed. To address this question, we examined the effect of the TIR length on the transposition reaction and determined that IstA requires the presence of two complete direct repeats in the pre-cleaved TIRs to perform the integration reaction (Fig. 1f). We also interrogated the role of the sequence specificity of the donor DNA in the transposition process (Fig. 1g). No integration product was detected when the TIR was replaced by a random sequence, even if it contained the terminal 3' CA dinucleotide that is normally cleaved by numerous DDE transposases. Similarly, no integration product could be detected when an unprocessed donor DNA bearing the flanking DNA was used (it had been previously suggested[36] that IstA is unable to cleave a linear donor DNA fragment). Interestingly, the presence of a flanking 5' overhang in pre-cleaved substrates stimulated the integration reaction (similar results have been described for MuA[47]), with an apparent maximal effect occurring at an extension of 4–5 nucleotides (Fig. 1g and Supplementary Fig. 2a). Collectively, our data show that a pre-cleaved full-length TIR with a short 5' overhang is the optimal substrate to promote efficient DNA integration in vitro.

### IstA engages the TIRs with high cooperativity to form a stable complex

The pronounced effect that the donor sequence and length had on the integration reaction led us to ask whether reduced IstA activity was caused by a decrease in DNA affinity. To test this idea, we used fluorescence anisotropy to assess the ability of IstA to recognize different nucleic acid substrates used in the integration assay. In particular, we examined long and short DNA molecules containing either random or specific sequences of the transposon end: i.e., the complete (R1-R2) and partial (R1) pre-cleaved right TIR, and two random sequences of similar size ($Rd_{60}$ and $Rd_{27}$). Since the isolated transposase is prone to precipitating in low salt conditions, we preformed this experiment with the protein fused to an MBP tag (the tagged protein shows the same levels of insertion activity as the wild-type protein (Supplementary Fig. 2b).

The affinity of the transposase for the full pre-cleaved transposon end (R1-R2) is relatively high, with an apparent $K_d$ of 125 nM (Fig. 2a). The affinity for the shorter DNAs and for the random sequences is similar or only slightly reduced (176 nM for R1, 191 nM for $Rd_{60}$ and 121 for $Rd_{27}$). Interestingly, however, IstA shows a stronger positive cooperativity for the sequence of the complete transposon end compared to the random sequences and shorter DNA fragments (Hill coefficient 3.6 for R1-R2 *vs.* 1.2 for R1, 1.5 for $Rd_{60}$ and 2.2 for $Rd_{27}$). Although it is not trivial to deconvolute this parameter in this case due to the likely presence of multiple species with different tumbling constants (1–4 protein subunits bound to 1–2 DNA fragments), the elevated value suggests that the transposase more readily adopts a conformational state necessary for self-assembly on its native nucleic acid sequence.

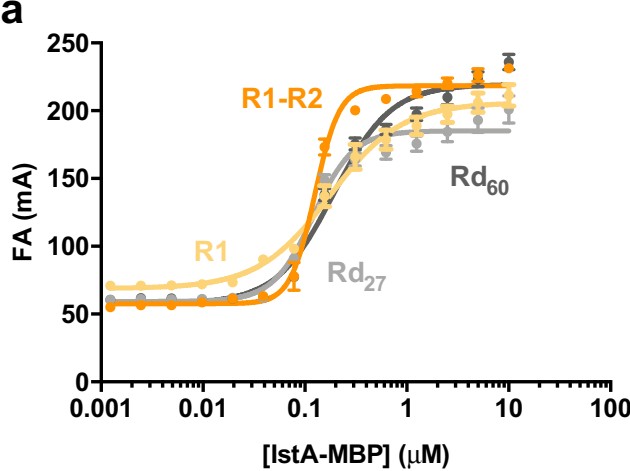

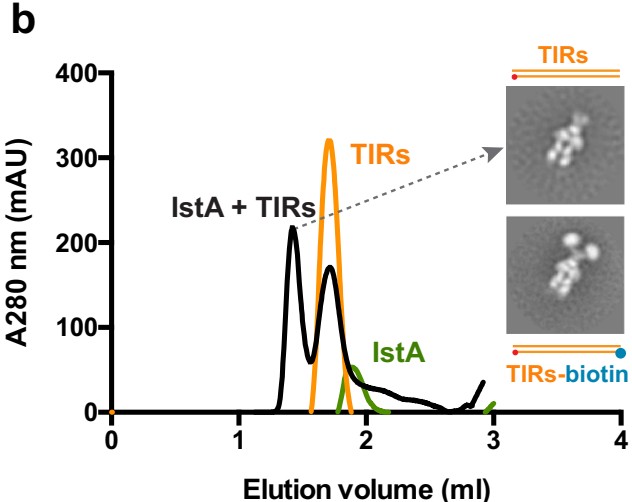

**Fig. 2 | DNA binding and oligomerization by IstA. a** Effect of sequence and length on the DNA binding activity of IstA as measured by fluorescence anisotropy (mA). R1-R2 (orange), pre-cleaved full right transposon end; R1 (yellow), half of the transposon end containing the first direct repeat; $Rd_{60}$ (dark grey), 60 bp-long random DNA; $Rd_{27}$ (light grey), 27 bp-long random DNA. Data points and error bars represent the mean values and standard deviation, respectively, between six independent experiments (except $Rd_{60}$ that was performed three times). Source data are provided as a Source Data file. **b** Analytical gel filtration shows that DNA-free IstA elutes as a monomer but readily forms a large molecular weight complex (~250 kDa) in the presence of appropriate transposon ends. Chromatogram of isolated IstA is shown as a green line, free TIRs in orange and the IstA•TIRs complex in black. Inset: Negative stain 2D class averages of the peak fractions containing IstA in complex either with the TIRs (upper class average), or with biotin-tagged TIRs preincubated with streptavidin (lower class average). Red dot indicates the position of the terminal CA dinucleotide that contains the donor 3′-OH while the blue dot shows the position of the biotin tag. Source data are provided as a Source Data file.

Intrigued by the high cooperativity, we next employed size exclusion chromatography to characterize the interaction between IstA and the transposon ends. Consistent with the results observed during the protein purification, IstA alone eluted as a monomer (Fig. 2b). However, in the presence of the pre-cleaved transposon ends, the transposase assembled into a stable high-molecular weight oligomer with an elongated shape and apparent 2-fold symmetry, as shown by negative staining electron microscopy (the gel filtration and fluorescence anisotropy assays performed with pre-cleaved donor DNAs gave identical results regardless of the presence of the overhang indicating that, although it appears to stimulate the strand transfer

reaction, the small 5′ extension does not have an appreciable effect on the affinity for the nucleic acid and the overall architecture of the complex) (Fig. 2b). Interestingly, when this experiment was performed using pre-cleaved TIRs tagged with biotin on one end, previously preincubated with streptavidin, the two-dimensional analysis revealed that the transposase complex is capable of engaging two transposon ends simultaneously (Fig. 2b). These findings support the idea that IstA specifically uses the sequence of its transposon ends to improve the efficiency of forming a stable, higher-order complex to bridge two donor DNA duplexes.

### IstA self-assembles into a tetramer to synapse the transposon ends

Transposases have been reported to self-assemble into dimers, tetramers and even octamers to synapse transposon ends[49–52]. The molecular architecture of these complexes, as well as the orientation and details of the interaction with the donor DNA, are strikingly diverse. To reveal the three-dimensional organization of IstA bound to the transposon ends, we turned to single-particle cryo-electron microscopy (cryo-EM). To prepare the complex, purified IstA was first incubated with the donor substrate. Since the integration activity of IstA bound to either an equimolar mixture of right and left ends or just the isolated right TIR was comparable, we collected preliminary negative-staining and cryo-EM datasets of the transposase bound to both substrates. In line with the similar levels of activity observed in the integration assay (Fig. 1e), the 2D averages and three-dimensional reconstructions were remarkably similar in both cases (Supplementary Fig. 3). However, the resolution and quality of the map of the complex of IstA bound to both transposon ends were lower, probably due to the presence of a mixture of left-right (~50%), left–left (~25%) and right–right (~25%) complexes. To minimize the heterogeneity we elected to use only the pre-cleaved right end for the structural studies (Supplementary Table 1), an approach widely used for other transposases[50,51,53,54]. IstA was then incubated with the optimal donor DNA in vitro (i.e., a complete pre-cleaved right transposon end with a 5 base-long 5′ overhang) and injected into a preparative gel filtration column. Peak fractions were applied to grids, vitrified, and imaged on a Titan Krios equipped with a Gatan K3 camera. The micrographs showed that the particles had good contrast and were well-distributed, while 2D class averages confirmed that the molecule can adopt different orientations on the grid (Supplementary Fig. 4). Interestingly, although the initial reconstructions indicated that the sample was monodisperse and homogeneous, they also revealed the presence of flexibility in certain areas, particularly at the apical and basal regions of the complex (Supplementary Fig. 4d). Despite this movement, subsequent rounds of 3D classification readily generated cryo-EM map with an estimated overall resolution of 3.4 Å (Supplementary Fig. 4 and Supplementary Table 2). The final reconstruction, which has an elongated shape of approximately 200 by 100 Å, showed good quality for both the protein and nucleic acid, allowing for the construction of a complete atomic model of the complex (Fig. 3a and Supplementary Fig. 5).

Overall, the cryo-EM structure shows that IstA self-assembles into a tetramer that engages two transposon DNA ends around a local two-fold symmetry axis (Fig. 3a). Close inspection reveals that the 400-residue long IS21 transposase is composed of four domains (Fig. 3b, c). The first of these elements are two N-terminal helix-turn-helix domains (HTH-1 and HTH-2, comprising amino acids 1–39 and 55–104, respectively), which are connected by a long minor groove-binding motif. Interestingly, this tandem arrangement has been reported for other transposases such as MuA (domains Iβ and Iγ), TnsB, Mos1, and Tc3, as well as the centromere-binding protein CENP-B, which probably evolved from an ancestral transposase[50,51,54–57]. A short linker, which confers the protein a significant degree of flexibility (Fig. 3c), separates these two modules from the catalytic DDE domain (residues 120–285). The DDE RNaseH-like fold (domain IIα in MuA) is shared with viral

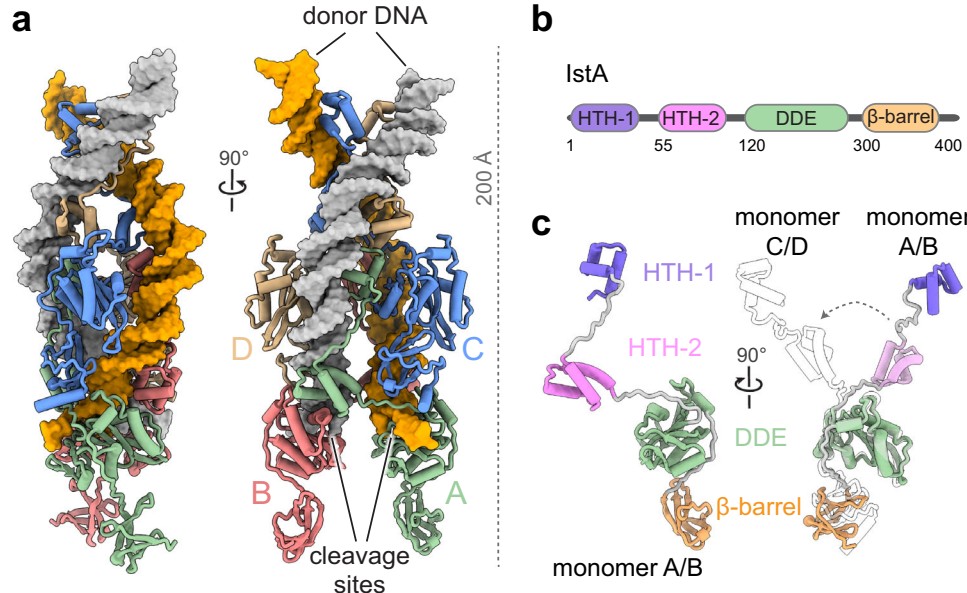

**Fig. 3 | Structure of IstA bound to IS21 transposon ends. a** Two orthogonal views of the IstA tetramer bound to two pre-cleaved TIR DNAs. **b** Domain organization of IstA. HTH helix-turn-helix domain, DDE catalytic domain. **c** The transposase oligomer comprises two monomer conformations. The upper (C, D) and lower (A, B) monomer pairs are mainly differentiated by a rotation between the second HTH fold and the DDE domain.

integrases and numerous prokaryotic and eukaryotic transposases[10], and is followed by a small ß-barrel (residues 300–355) similar to domain IIß found in MuA[50,58]. Although the last 45 amino acids of IstA are predicted to form a flexible loop containing two α-helices, a short helical fragment of this C-terminal extension could be modelled in only two of the molecules (chains C and D).

Collectively, the four transposase molecules form a highly interdigitated network in which each monomer contacts all three partner subunits (Figs. 3a and 4). Dimerization of the upper monomers (chains C and D) is mainly mediated by equivalent domains of symmetry-related molecules (HTH-2$_C$/HTH-2$_D$ and DDE$_C$/DDE$_D$) (Fig. 4a). In contrast, the lower subunits (chains A and B) are bridged by a domain-swapped configuration whereby the second DNA binding domain engages the DDE module of the opposite monomer (HTH-2$_A$/DDE$_B$ and HTH-2$_B$/DDE$_A$) (Fig. 4a). The interaction between the top and bottom dimers is mediated by two key areas (Fig. 4b). In one, the ß-barrel of the upper subunits rest atop of the RNaseH-like modules of the lower monomers (ß-barrel$_C$/DDE$_A$ and ß-barrel$_D$/DDE$_B$). In the other, more central region of the complex, the HTH-1 domains of the lower subunits form extensive contacts with one HTH-2 (HTH-1$_A$/HTH-2$_D$ and HTH-1$_B$/HTH-2$_C$) and two DDE domains (DDE$_C$/HTH-1$_A$/DDE$_D$ and DDE$_C$/HTH-1$_B$/DDE$_D$) of the upper molecules. Interestingly, it has been reported that IstA can be also expressed as an alternate isoform, termed a cointegrase, which lacks the first 8 residues located in this central interaction area. This IstA variant is reported to evince a more specialized functionality that can process IS21 tandems with higher efficiency than the full-length transposase (Supplementary Fig. 1b)[37]. Some of these N-terminal residues are located at the core of the multifaceted interface (Fig. 4b), suggesting that their absence likely affects the global stability and structure of the complex.

## Differential binding of IstA molecules regulates DNA transposition

Although the IstA subunits are highly interconnected, there is a clear division of labor between them. The uppermost dimers engage the TIR sequences in *cis* and appear to play a mostly structural function (Fig. 5a): the DDE domains in these subunits do not interact with the DNA and the catalytic triad is far (>8.5 Å) from the nucleic acid (the

loop present immediately after the catalytic E233 is additionally disordered) (Fig. 5b). In contrast, the lowermost dimers engage the transposon ends in *trans* (i.e., the DDE folds engage a different DNA end than the segment bound by their HTH elements) and appear most likely to perform a catalytic role (Fig. 5c). This particular arrangement has been reported for numerous transposases and serves as a critical checkpoint to prevent the formation of DNA breaks before the two transposon ends are bridged together[49–51,53,54,57].

The DDE motifs (D124, D187 and E233) of the catalytic monomers are proximal to the signature 3' CA dinucleotide at the end of the donor DNAs and thus are correctly positioned to perform a transposition reaction (Fig. 5d). Mutation of the catalytic residues results in an almost complete abrogation of the integration activity (Fig. 5e and Supplementary Fig. 6). Consistent with the known metal dependency of the catalytic reaction, extra density was observed close to the 3'-OH of the cleaved deoxyadenosine that could be modelled as a Mg$^{2+}$ ion (coordinated with D124 and E233). The presence of only one divalent metal in the active site has been reported previously for other transposases, and may be due to the moderate concentration of magnesium (5 mM) in the cryo-EM buffer[51,54] or to the fact that the complex represents a post-cleavage intermediate of donor DNA. The loop close to the catalytic glutamate (residues 225–228) that is flexible in the upper monomers becomes ordered in the catalytic subunits, where can be seen to act as a wedge that induces a base flip in the nucleotide complementary to the 3' deoxyadenosine, which is itself stabilized by interactions with a tyrosine (Tyr 113), positioned in the loop that connects the DDE and HTH-2 domains (Fig. 5d). Additionally, a glutamine (Gln 369) located in the C-terminal region of the upper transposase molecule appears to make polar contacts with the flipped base (Fig. 5d). Mutation of this residue significantly reduces the integration activity (Fig. 5e and Supplementary Fig. 6). This mechanism is reminiscent of that seen in other systems such as the RAG1-RAG2 recombinase, the recently determined TnsB strand transfer complex, and the PFV and SIVrcm intasomes (Supplementary Fig. 7)[54,57,59–61]. All these enzymes position the bases of the 5' non-transferred overhang in a similar flipped configuration, possibly as a regulatory mechanism to ensure correct formation of an active synaptic complex.

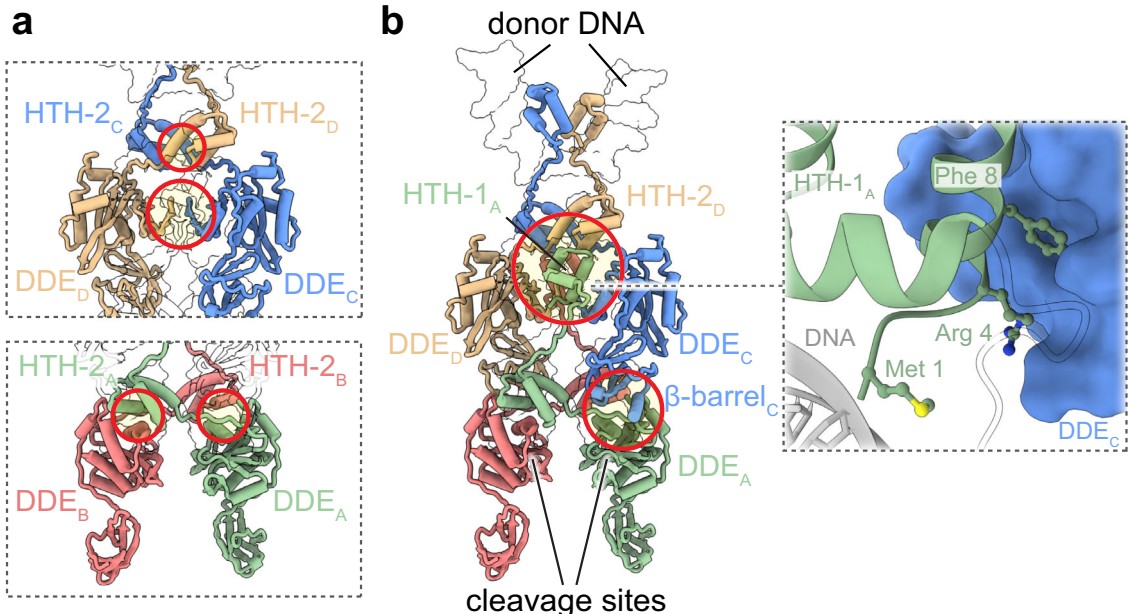

**Fig. 4 | Protein–protein interactions in the IstA tetramer. a** The main interaction areas between the upper (HTH-2/HTH-2 and DDE/DDE of chains C and D, top panel) and lower (HTH/DDE of chains A and B, bottom panel) transposase monomers are highlighted with red circles. **b** Interactions between the upper and lower IstA monomers occur between HTH-1 of one lower chain and HTH-2/DDE of both upper chains, and interactions between the ß-barrels and the integrase domains (two of the symmetry-related interactions are not highlighted for clarity). Detail of the contacts of the first eight residues that are absent in an IstA variant that possesses high intrinsic cointegrase activity (right inset).

## IstA recognizes the transposon ends to form an intertwined plectoneme

Transposases must recognize donor transposon ends with high accuracy to prevent nonspecific DNA breaks and to ensure that the length of the mobile element is not altered after every cleavage and insertion cycle. Notably, the division of labor of the two IstA dimers also extends to DNA recognition. The reconstruction reveals, in agreement with our observation showing that IstA requires the presence of a complete TIR to perform efficient DNA transposition (Fig. 1f), that the catalytic subunits engage the R1 direct repeat whereas the upper ones interact with the R2 sequence (Fig. 6).

Close analysis reveals that, although the TIR sequences are very similar, they are not identical (Supplementary Fig. 8a). More precisely, sequence alignment shows that each direct repeat contains two highly conserved GC-rich regions (CS1 and CS2) separated by ~10 more variable base pairs (i.e., one complete DNA turn), which are mostly composed of AT nucleotides (Fig. 6a). Interestingly, the HTH-1 domains of the catalytic and structural monomers make specific contacts with the CS2 regions of the R1 and R2 repeats, respectively, while the HTH-2 modules establish similar interactions with the CS1 sequences. (Fig. 6b, c and Supplementary Figs. 8b, 9). Several amino acids (e.g., R31, K32, T92 and K95) recognize the CS sequences through major groove interactions. Mutation in some of these residues significantly reduced integration activity (Fig. 6d). By comparison, the less ordered linker that connects these domains makes fewer contacts with the minor groove of the AT-rich region (Fig. 6b, c).

The interactions between the transposase molecules define a three-dimensional architecture that drastically remodels the donor DNA. Each duplex is sharply bent by ~70° (Fig. 6b), with a specially curved area close to the R1-R2 junction that appears triggered by the interaction with the HTH-1 domains of the catalytic subunits. Interestingly, the pair of transposon ends are positioned so as to mimic two right-handed DNA crossings – one at the active site and another at the R2 interaction – consistent with previous predictions that suggested that the spacing observed between direct repeats (separated by 23 bp) could favor the formation of supercoiled DNA structures (Fig. 6e)[34,62,63].

Overall, the IS*21* transposase appears to specifically recognize a pair of donor DNA molecules that wrap around the IstA tetramer in a manner consistent with that adopted in a negatively supercoiled substrate.

## Discussion

Transposons are ubiquitous mobile genetic elements found in prokaryotic and eukaryotic organisms that play a key role in horizontal gene transfer, genome organization and evolution[2]. Although numerous transposon elements have been widely studied during the last decades using genetic and biochemical methods, the structural and mechanistic understating of these systems is still limited. Interestingly, the available atomic models revealed a striking variability in the three-dimensional organization of the transposase molecules and the path of the substrate DNAs, highlighting that there exist complex levels of molecular regulation for transposition events. Here, we use a transposase of the widespread IS*21* family to help to understand the recognition of the often-multiple terminal transposon repeats and set the stage for molecular analyses of nucleotide-dependent DNA transposition.

Although the biochemical activity of some IS*21* family members had been analyzed previously using experiments conducted with cell lysates and in vivo[36,37,42], the absence of any study performed with fully purified components has hampered an understanding of their mechanism. By reconstituting an in vitro assay for this system, we were able to confirm that both gene products of IS*21*, IstA and IstB, are essential for transposition and that the transposase can use either the isolated left and right TIR or a mixture of the two TIR sequences to perform insertion reactions (Fig. 1e). Using a pre-cleaved substrate – which has been shown to stimulate transposition in other systems[64,65] – we further found that the full-length TIR sequence with a short overhang is the optimal substrate for IS*21* transposition (Fig. 1d–g).

The in vitro assay data allowed us to identify ideal donor DNA substrates for biophysical and cryo-EM analysis. These studies in turn revealed that IstA uses the TIR substrate to promote its highly cooperative assembly into a tetrameric complex that engages two transposon ends (Figs. 2 and 3). Interestingly, the protein-nucleic acid

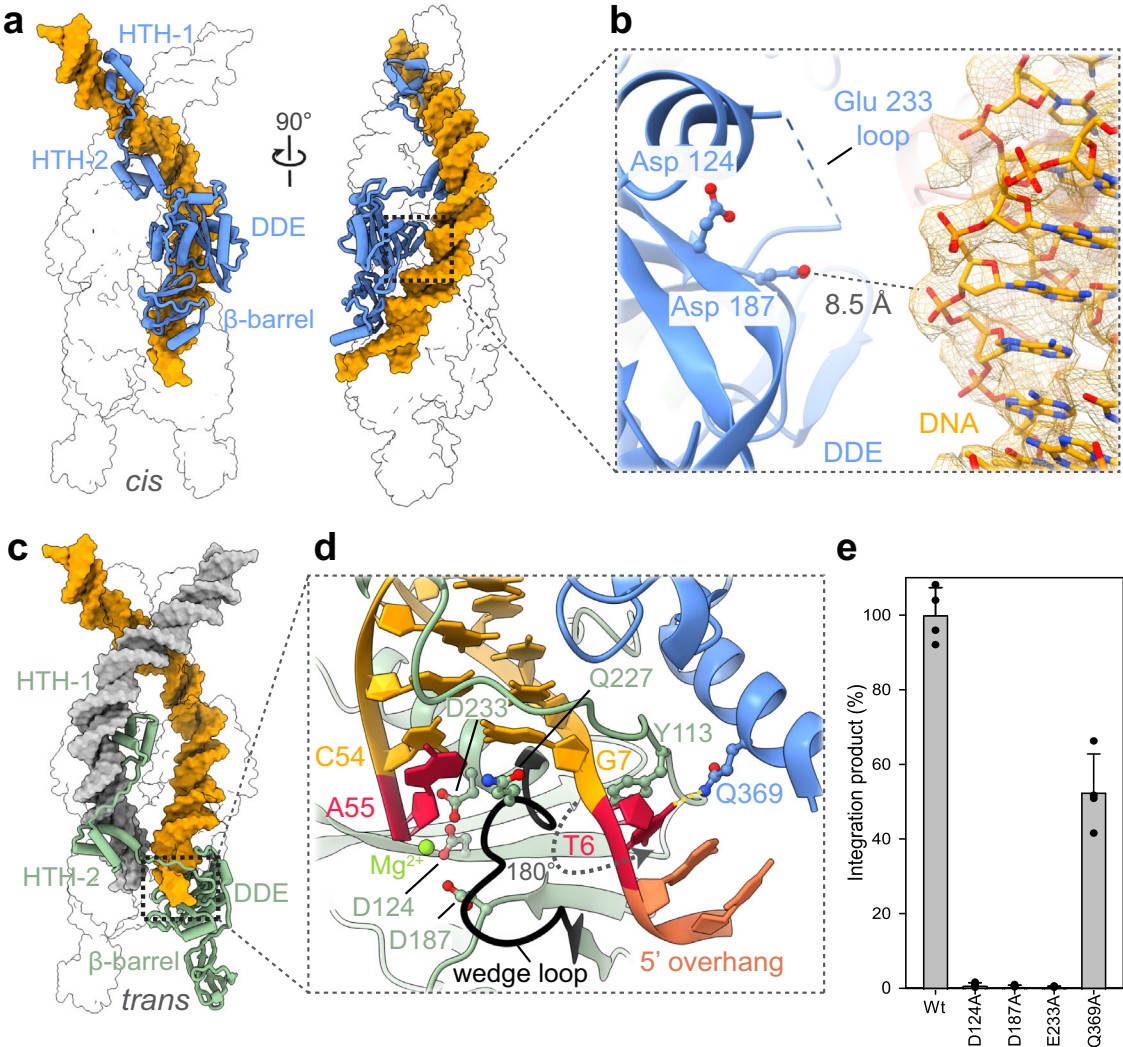

**Fig. 5 | The IstA tetramer is formed by two structural and two catalytic subunits. a** Two orthogonal views of one of the upper monomers (chain C) binding the transposon end in *cis*. Only one transposase molecule is shown for clarity. **b** Close-up view of the boxed area highlighted in (**a**). Density of the DNA is contoured at a threshold value of 0.0245. Two acidic catalytic amino acids of the DDE fold are shown as ball-and-stick. **c** The catalytic monomers interact with DNA ends in *trans*. Only chain A is shown for clarity. **d** Detailed view of the IstA active site. Chains A and C are shown (green and blue, respectively). The DDE motif and some residues critical for the interaction with the DNA are depicted as balls and sticks. A well-defined 'wedge' loop (black) inserts between the terminal adenine and its complementary thymine (both colored in red), flipping the conformation of this base and the two nucleotides visible in the 5 bp-long overhang (salmon). **e** Alanine mutants of residues implicated in catalysis and recognition of the 5' overhang have a significant loss in integration activity. Integration activity identified as appearance of linear DNA product relative to the WT protein. Boxes and error bars represent the mean value and standard deviation, respectively, between four independent experiments (except E233A that was repeated three times). Dots indicate the individual values. Source data are provided as a Source Data file.

interactions and parallel configuration of the donor DNAs observed here are reminiscent to those seen in the Tc1/mariner member Mos1, which bears an HTH domain that also plays a critical dual role mediating transposase oligomerization and transposon end synapsis[51] (Fig. 7a). Mos1, however, acts as a dimeric transposase, likely because of its shorter transposon ends (which only contain one terminal repeat). Mos1 also lacks a C-terminal β-barrel that appears to play an important role in stabilizing the IstA tetramer.

In addition, the IS*21* transposase bears striking similarities with the tetrameric structure of TnsB, recently determined in complex with a strand-transfer substrate (Fig. 7a)[54,57]. The reconstructions of the Tn7-like transposase show a comparable domain organization and reveal that TnsB self-assembles establishing an extensive interaction network. The first HTH domain (termed Iβ and NTD1 in TnsB), which also lies at the core of the tetramer, contains a small N-terminal extension that makes contacts with the neighboring subunits, highlighting the role of this N-terminal domains in the stabilization of the synaptic

complexes (Fig. 4b and Supplementary Fig. 10). Interestingly, although the overall architecture of the tetramers is remarkably similar, the HTH tandems of the non-catalytic subunits of TnsB and the cognate L2 and R2 terminal repeats of the donor DNA molecules were not resolved in the cryo-EM reconstructions (Fig. 7a)[54,57]. How TnsB recognizes the additional terminal repeats present in Tn7 elements in the context of the transposome and what is their precise function, therefore, remain an open question.

One of the best-studied close homologs of IstA, however, is the bacteriophage MuA transposase. Although Mu possesses auxiliary protein and nucleic acid elements that give this system additional levels of regulation, the core domains and DNA sequences that are essential to perform the transposition reaction – i.e., tandem DNA binding motifs (Iβ and Iγ in MuA), DDE (IIα), β-barrel domain (IIβ) followed by an alpha-helical extension (IIIα), together with two transposon ends containing multiple direct repeats – are remarkably analogous to those required for IstA (Supplementary Fig. 10)[47]. Despite

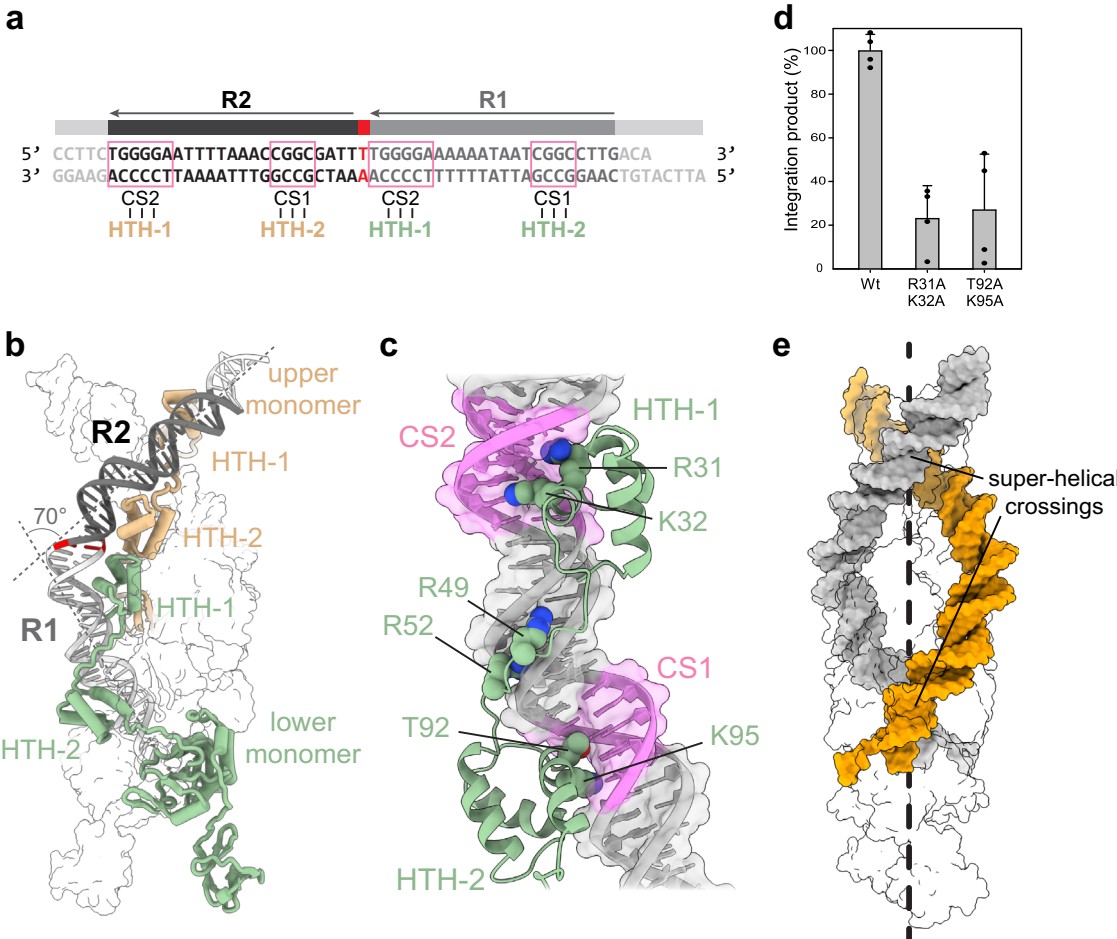

**Fig. 6 | IstA bends and twists two transposon ends into a negative superhelix.**
**a** Details of the right transposon end sequence used for the structural analysis.
**b** The two catalytic monomers of the IstA tetramer (chains A and B) interact with the R1 repeat or a TIR while the upper structural subunits (chains C and D) engage the R2 sequence. Half of the complex is shown as a transparent surface for clarity.
**c** The HTH domains of the lower and upper monomers use the same residues to interact with equivalent nucleotides in the R1 and R2 repeats. **d** Double alanine

mutants of HTH residues implicated in DNA binding have a significant effect on integration activity. Integration activity identified as appearance of linear DNA product relative to the WT protein. Boxes and error bars represent the mean value and standard deviation, respectively, between four independent experiments. Source data are provided as a Source Data file. **e** The donor DNA ends are twisted into two super-helical crossings. IstA tetramer shown as a transparent surface for clarity.

this similarity, the three-dimensional arrangement of the two transposases initially appears strikingly divergent. For example, IstA recognizes the donor DNA in a clear, negatively supercoiled configuration, whereas MuA engages the transposon ends in a splayed-out conformation in the post-transposition state (Fig. 7b)[50]. However, despite this difference, it has been reported that MuA may also adopt an intertwined topological state, in which the L2 and R2 sequences would sit on top of the R1 and L1 repeats respectively[66,67], forming two superhelical crossings[67]. Although Mu's left end is more complex than the one found in IS*21* (L1 and L2 are separated by a longer spacer sequence that contains an HU binding site), the structure of the IS*21* transposase is remarkably consistent with these topological predictions, suggesting that MuA may adopt a similar configuration at a different stage (pre-strand transfer) of the transposition reaction, and highlighting the plasticity and complexity of the regulation of this type of enzymes.

The results presented here, in addition, provide insights into IS*21* regulation. DNA transposition is tightly controlled to prevent DNA damage and recombination events. As such, the process is controlled at many levels[2,68]. Transposases, for instance, are often produced as inactive monomers that require the recognition of the transposon ends to assemble into active multimers to perform the strand transfer reaction in *trans*. In line with this behavior, IstA is expressed as an

isolated monomer but readily forms a tetramer in the presence of the donor DNA (Fig. 2b). This regulated assembly mechanism has been observed for other transposases, such as MuA, and ensures that the active transposase is only assembled on the transposon ends when the conditions required for a productive transposition reaction are met[58,69,70].

The transposon ends of numerous families of mobile elements frequently contain endogenous promoters to control transposase expression[68]. The presence of these promoters has important ramifications for a host organism, as the insertion of such elements can locally affect the expression of neighboring sequences, including antibiotic resistance genes[24,71]. IS*21*, in particular, contains a −10 box and −35 box in its left and right TIRs, respectively (Fig. 8). As such, the promoter activity of a single transposable element is relatively weak[35]; however, when IS*21* forms tandem arrays, the −35 sequence becomes positioned with the correct spacing upstream of the −10 box, forming a relatively strong promoter with increased transposition activity[35]. Interestingly, our results show that IstA binding occludes the promoter regions of the transposon ends, suggesting that TIR sequestration by the transposase autoregulates expression levels in the ~2 kb operon (Fig. 8).

In addition to showing that IstA can self-assemble into a tetramer that specifically synapses two negative supercoiled transposon ends in *trans*, the state of the transposase imaged here appears to be correctly

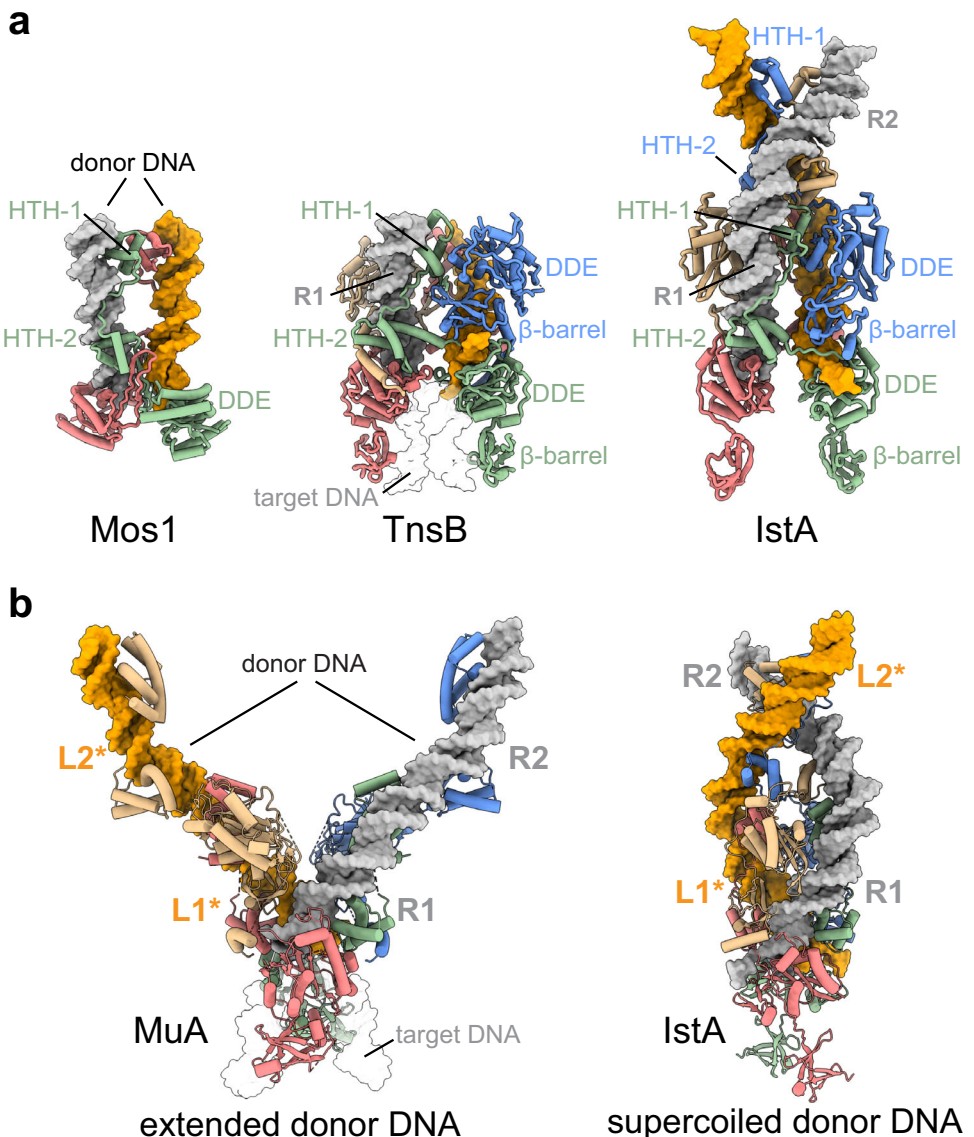

**Fig. 7 | Comparison between IstA, Mos1, TnsB and MuA transposases. a** Similar views of Mos1 (PDB ID 3HOS [https://doi.org/10.2210/pdb3HOS/pdb]), TnsB (PDB ID 8AA5 [https://doi.org/10.2210/pdb8AA5/pdb]) and IstA (from the two highly-similar TnsB tetrameric structures, the one obtained at higher resolution has been used to perform the structural comparisons). Each transposase monomer has been colored differently. Donor DNA monomers colored in orange and grey. TnsB target DNA is shown as transparent surface for clarity. **b** Post-strand transfer MuA (PDB ID 4FCY [https://doi.org/10.2210/pdb4FCY/pdb]) and pre-strand transfer IstA tetramers adopt very different conformations. The orientation of IstA is orthogonal to that shown in panel (**a**). The target DNA in the MuA structure is shown as transparent surface for clarity. Asterisks in the L1 and L2 repeats indicate that both structures have been determined with two right transposon ends.

poised to perform a strand transfer reaction (Fig. 5d). However, we and others have shown that IstA is inactive in the absence of its regulatory factor, the IstB ATPase (Fig. 1d). Several classic mobile elements, including Tn7 and bacteriophage Mu, also depend on ATP-dependent factors to select the appropriate insertion sites[70,72–74]. Recently, the structures of a recruitment complex and a holo-transposome from a Tn7-like element have been reported[75,76]. Although these studies help to define how the AAA + factor selects the insertion site, the structure of the TnsB tetramer remains largely unchanged and, therefore, how these 'molecular matchmakers' precisely regulate the transposase activity to promote DNA transposition remains unclear. Interestingly, detailed comparison of the active sites reveals that the spacing between the catalytic DDE motifs of MuA and TnsB is larger than the one seen in IstA (-44 Å and 40 Å vs 31 Å, respectively) (Supplementary Fig. 11)[50,54,57]. Modelling of either an intact target nucleic acid duplex or the highly distorted strand transfer substrates present in the MuA and TnsB structures into the active site of IstA shows that the geometry of

the synaptic complex observed here could generate two staggered cuts that would produce shorter directs repeats than the ones observed experimentally (2–3 vs. 5 bp) (Supplementary Fig. 11). All these analysis, together with the observation that IstB is required for the formation of correct target duplications[42], indicates that the IS21 ATPase likely participates in the alignment of the target sequence into the active site by remodeling the nucleic acid (to a different degree that the one described for MuA and TnsB) and/or by altering the configuration of IstA. Future experiments involving IstA and IstB will be needed to test these ideas.

## Methods
### Protein expression and purification
*Geobacillus stearothermophilus* IstA was purified as described previously with minor modifications[43]. The DNA fragment encoding the full-length transposase sequence was cloned into a pET derived vector with a TEV cleavable MBP-His6 tag at the C-terminus (2CcT vector) and

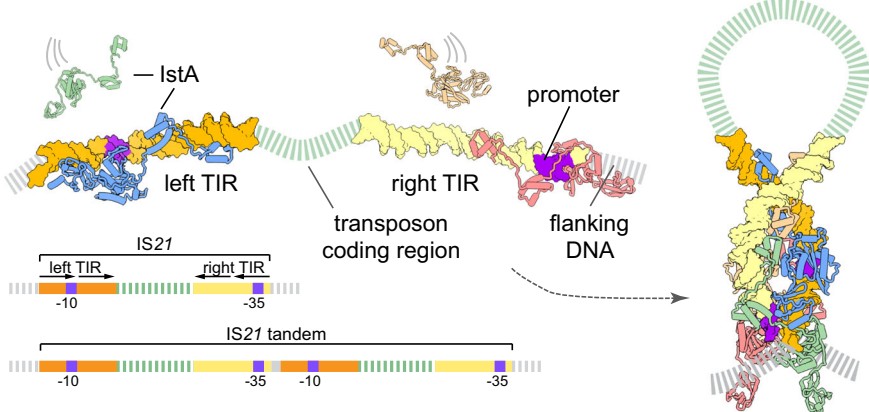

**Fig. 8 | Model for IstA assembly and donor DNA recognition.** Proposed assembly model for how IstA binding regulates the expression of the IS*21* operon by masking promoter elements in the donor transposon ends.

was then transformed into a C41 (DE3) *E. coli* strain (Lucigen). Cells were grown at 37 °C in 2xYT (Yeast Extract Tryptone) medium supplemented with 1% glucose, until optical density at 600 nm reached a value of 0.7, at which point β-isopropyl-D-1-thiogalactopyranoside (IPTG) was added to a final concentration of 0.5 mM to induce the expression during 4 h. Cells were centrifugated, resuspended in lysis buffer [50 mM HEPES pH 7.5, 750 mM NaCl, 5 mM MgCl₂, 5% glycerol and 1 mM β-mercaptoethanol (β-MeOH)] supplemented with protease inhibitors (cOmplete, Mini Protease Inhibitor Tablets; Roche) and lysed by sonication. The lysate was clarified by centrifugation (20,000 g, 30 min., 4 °C) and the supernatant was filtered through 0.45 μm pore-size filter (Sartorius). The sample was bound to a Ni-Sepharose column (5 ml His-Trap HP Chelating; Cytiva), washed first with 8 column volumes (CV) of lysis buffer containing 30 mM imidazole and next equilibrated with 5 CV of the same buffer with 150 mM NaCl. Ni-column was eluted directly onto a cation exchange column (5 ml Hi-Trap SP HP column; Cytiva) by applying 6 CV of 50 mM HEPES pH 7.5, 150 mM NaCl, 5 mM MgCl₂, 5% glycerol, 300 mM imidazole and 1 mM β-MeOH. The ion exchange column was further washed with 6 CV of 50 mM HEPES pH 7.5, 300 mM NaCl, 5 mM MgCl₂, 5% glycerol and 1 mM β-MeOH, and the sample was eluted using a 13 CV linear gradient of salt reaching a maximum NaCl concentration of 1 M. Peak fractions were incubated overnight at 4 °C with TEV protease. For removal of MBP tag, the sample was applied to a second histidine-affinity purification step (5 ml His-Trap HP column, Cytiva), where the retained cleaved protein was eluted by applying 13 CV of a linear gradient of the sample elution buffer containing 300 mM imidazole and next concentrated with ultrafiltration devices (Amicon Ultra-15, 10 kDa MWCO cutoff; Millipore), and further purified by gel filtration loading the concentrated sample into a preparative HiPrep Sephacryl S-200 16/60 HR column (Cytiva) equilibrated in 50 mM HEPES pH 7.5, 750 mM NaCl, 5 mM MgCl₂, 5% glycerol and 1 mM β-MeOH, at room temperature. IstA fractions were pooled and concentrated up to 23 μM as above, flash-cooled in aliquots in liquid nitrogen and stored at −80 °C until use. IstA-MBP was prepared as the untagged version, except that the TEV proteolysis and ortho-Ni steps were omitted. IstA mutant constructs were generated using the round-the-horn protocol and expressed and purified as the wild-type protein.

*G. stearothermophilus* full-length IstB was purified as described previously[43]. Briefly, His6-MBP-IstB was expressed using a pET derived vector in *E. coli* BL21 codon-plus (DE3) RIL cells (Stratagene). Cells were grown at 37 °C in 2xYT media, induced at an A600 of 0.6 with 0.5 mM IPTG at 37 °C for 3 h, harvested by centrifugation, resuspended in resuspension buffer (20 mM HEPES pH 7.5, 500 mM NaCl, 5 mM MgCl₂, 5% glycerol, 30 mM imidazole, 1 mM β-MeOH and 0.1 mM ADP) supplemented with protease inhibitors (cOmplete, Mini Protease Inhibitor

Tablets; Roche), and lysed by sonication. The supernatant obtained after centrifugation (20,000 g, 30 min., 4 °C) was filtered through 0.45 μm pore-size filter (Sartorius) to be run over a Ni-Sepharose column (His-Trap HP, Cytiva) and washed with resuspension buffer. The protein was then eluted with 500 mM imidazole buffer onto an amylose affinity resin and cleaved on-column overnight at 4 °C with PreScission-protease. After elution by passing resuspension buffer, the protein was concentrated with ultrafiltration devices and injected in a HiPrep Sephacryl S-200 16/60 HR gel-filtration column (Cytiva) pre-equilibrated in 20 mM HEPES pH 7.5, 500 mM NaCl, 5 mM MgCl₂, 5% glycerol, 0.1 mM ADP and 1 mM β-MeOH at room temperature. Fractions corresponding to IstB were pooled and concentrated, flash-cooled in aliquots in liquid nitrogen and stored at −80 °C until use.

**In vitro integration assay**

Integration reactions were performed in 50 mM HEPES pH 7.5, 150 mM NaCl, 5 mM MgCl₂, 5% glycerol, 0.05 mg/ml BSA and 1 mM ATP. IstA or IstA-MBP was pre-incubated with short linear donor DNA molecules as indicated (both present at 1 μM in the final reaction) (Supplementary Table 1). IstB (1 μM unless indicated otherwise) was independently pre-incubated with supercoiled pSG483, a derivative of pUC19, which was used as target DNA (10 nM final concentration). Both proteins were then mixed in 30 μl and incubated 30 min at 37 °C. Reactions were quenched by incubating 20 min at 37 °C with proteinase K (0.25 mg/ml final concentration), SDS (1% final concentration) and EDTA (28 mM final concentration). Samples were run for 18 h on 1.5% (w/v) TAE agarose gels (40 mM sodium acetate, 50 mM Tris–HCl pH 7.9, 1 mM EDTA pH 8.0) at 2–2.5 volts/cm. To visualize the DNA, gels were stained with 0.5 μg/ml ethidium bromide in TAE buffer for 20 min, destained in TAE buffer for 30 min, and exposed to UV transillumination. Images of representative gels can be seen in Fig. 1, Supplementary Information and Source Data file.

The products of the insertion reaction consist on linearized target plasmids flanked by direct repeats, generated as a consequence of the strand-transfer reaction, and one TIR on each end. To sequence these products, the reaction was performed using a donor DNA with a phosphorylated overhang mimicking a digested XhoI sequence at the internal end of the TIR (opposed to the reactive CA). To avoid missing the insertions that could happen in the AmpR gene, the product of 8 reactions was pooled, ethanol precipitated and ligated with a PCR fragment containing the KanR gene flanked by XhoI sites that had been previously digested with the restriction enzyme. The ligation positions the TIRs and direct repeats flanking the KanR gene. The re-circularized plasmids were transformed into XL1 blue competent cells (Agilent) and plated into KanR plates. The plasmid of ten colonies, containing the reaction product, was extracted using a miniprep kit (Macherey-Nagel)

and subjected to DNA Sanger sequencing using primers that annealed with the ends of the KanR gene, giving an accurate reading of the adjacent TIRs and direct repeats.

### DNA binding assays

0–10 μM IstA-MBP (in 2-fold dilution series) was titrated into reactions (20 μl final vol) containing 50 mM HEPES pH 7.5, 150 mM NaCl, 5 mM MgCl$_2$, 10% glycerol, 1 mM β-MeOH, 0.1 mg/ml BSA and 5 nM of TIR derived or random 5′-FAM-labeled dsDNA substrates (Supplementary Table 1). Fluorescence polarization measurements were performed at 37 °C on a Tecan Spark microplate reader using 384-well plates after mixing and incubating for 5 min. Data were converted to fluorescent anisotropy and fitted to a Hill binding equation:

$$Y = Bmax*X^{\wedge}h/(Kd^{\wedge}h + X^{\wedge}h) + Background$$

where Y is fluorescent anisotropy (mA), X is protein concentration (μM), Bmax is the maximum extrapolated binding in the same units as Y, Kd is the ligand concentration needed to achieve a half-maximum binding at equilibrium (expressed in the same units as X), h is the Hill slope and Background is the binding measured in wells that contain no added protein. Data analysis and representation were carried out using GraphPad Prism 7.

### Analytical size exclusion chromatography

To analyze the oligomeric state of IstA the transposase was mixed in a 1:1.5 (protein:DNA) molar ratio with the a stoichiometric mixture of left and right TIRs and dialyzed overnight at 4 °C into 50 mM HEPES (pH 7.5), 150 mM NaCl, 5 mM MgCl$_2$, 5% glycerol and 1 mM DTT. On hundred microliters of each sample at 8 μM were then run over a Superdex 200 5/150 GL analytical gel filtration column (Cytiva) in the dialysis buffer (except for the isolated transposase that was run in the same buffer but with 750 mM NaCl to avoid protein precipitation in the absence of nucleic acid- and monitored by A280 at room temperature). For the analysis of the complex with biotin-tagged TIRs the experiment was performed using the same conditions except that the DNA was previously incubated with streptavidin in a (1:1) molar ratio.

### Sample preparation for high-resolution cryo-electron microscopy

To obtain the IstA/right-TIR complex sample to perform cryo-EM experiments, IstA was mixed in a 1:1.5 (protein:DNA) molar ratio with the right TIR (Supplementary Table 1), containing a 5-nt 5′ overhang, and dialyzed overnight at 4 °C against 50 mM HEPES (pH 7.5), 150 mM NaCl, 5 MgCl$_2$, 5 % glycerol and 1 mM DTT. The sample was loaded onto a HiPrep™ Sephacryl S-200 16/60 HR column (Cytiva) equilibrated in dialysis buffer, at room temperature. The fractions corresponding to the peak containing IstA-DNA complex (~42 ml) were pooled and concentrated to 2.45 mg/ml (51.5 μM) using Amicon ultrafiltration devices (10 kDa cutoff; Millipore), flash-frozen with liquid nitrogen and stored at −80 °C.

For the cryo-EM experiments, the sample was diluted in 50 mM HEPES pH 7.5, 150 mM NaCl, 5 mM MgCl$_2$, 1 mM DTT and 0.005% Tween-20 to improve the particle distribution and angular coverage. 4 μl of the IstA-DNA complex (2.5 μM) was applied to glow-discharged C-Flat holey grids (CF-1.2/1.3; 400 mesh), blotted for 2 s (blot force 0) and frozen in liquid ethane using a Vitrobot Mark IV plunging system (ThermoFisher Scientific).

### Cryo-electron microscopy data acquisition

Cryo-EM grids were pre-screened in a JEOL 1230 microscope equipped with a TemCam-F416. Preliminary cryo-EM dataset of the IstA complex with the isolated right TIR and equimolar mixture of right and left TIRs were collected on a Talos Artica/Falcon III and a Glacios/Falcon IV microscopes (ThermoFisher Scientific), respectively. High-resolution data of the IstA complex with the isolated right TIR were collected on a Titan Krios electron microscope operated at 300 kV (Diamond Light Source). Imaging was performed using EPU at a nominal magnification of 81,000 x (calibrated physical pixel size 1.06 Å/px; super-resolution 0.53 Å/px) using a Gatan K3 camera direct electron detector operating in super-resolution counting mode. Each movie was recorded during 3.36 s in 50 frames with a nominal defocus range of −2.7 to −1.2 μm (increments of 0.3 μm). The dose rate was 1.19 e$^-$/Å$^2$/frame, resulting in an accumulated exposure of 59.7 e$^-$/Å$^2$ (Supplementary Table 2).

### High-resolution image processing

7215 movies were imported into RELION-3.1[77,78], motion corrected and electron-dose weighted with MOTIOCOR2[79], and the contrast transfer function (CTF) was estimated using GCTF[80]. A subset of the micrographs was picked with GAUTOMATCH, binned by two, and subjected to 2D classification. The resulting 2D averages were then used as templates to pick an entire dataset with GAUTOMATCH. A total of 11,094,653 particles were extracted and down-sampled to 3.18 Å/px. After 2D classification, 4,548,336 particles were selected and subjected to 3D classification, using C2 symmetry and a tight mask, which yielded a good quality class containing 1,101,138 particles (Supplementary Fig. 4). These particles were extracted with the original pixel size of 1.06 Å/px and used as input for a subsequent 3D refinement run, using a soft-edged mask that followed the contour of the particle, resulting in a 4 Å-resolution map. Although the core of the complex is quite stable, the upper and lower regions showed a significant degree of flexibility. The particles, thus, where subjected to a second round of 3D classification, after performing CTF refinement and Bayesian polishing, that allow us to select a subset of 337,376 particles that were used to generate the final 3.4 Å resolution map.

### Model building and refinement

Homology models for IstA were generated first with SWISS-MODEL server and later with ALPHAFOLD[81,82], and used as starting points to manually build the asymmetric unit of the protein (chains A and C) in COOT[83]. GRAPHITE-LIFE EXPLORER[84] was used to generate the DNA duplex (chains E and F). The complete asymmetric unit was then subjected to iterative rounds of model building and real-space refinement with COOT and PHENIX[83,85] using Ramachandran, rotamer, geometry and secondary structure restraints for the protein, and base-pair parallel planes, stacking and hydrogen bonds restraints for the DNA (generated using LIBG and ProSMART tools from CCP4 platform). The tetrameric conformation was obtained after applying C2 symmetry operators to the asymmetric unit. The quality of the map allowed to model and real-space refine almost the complete sequence of the four monomers, except for the beta-barrel of the lower subunits (chains A and B) that showed an increased degree of flexibility and was rigid-body fitted using the unsharpened map as reference. The final model was achieved by consecutive rounds of real space refinement performed with PHENIX applying NCS constraints. The quality of the model was assessed with MolProbity from Phenix package, the validation tool from PDB (OneDep) and different validation tools from Coot. All structure figures were generated with ChimeraX 1.4[86].

### Reporting summary

Further information on research design is available in the Nature Portfolio Reporting Summary linked to this article.

## Data availability

The cryo-EM densities and atomic coordinates of IstA bound to the transposon end generated in this study have been deposited in the EMBD and PDB under the accession codes EMD-15848 [https://www.

ebi.ac.uk/pdbe/entry/emdb/EMD-15848] and 8B4H [https://doi.org/10.2210/pdb8B4H/pdb], respectively. The PDB codes of the previously determined structures used in this manuscript are: 3HOS [https://doi.org/10.2210/pdb3HOS/pdb] (Mos1), 8AA5 [https://doi.org/10.2210/pdb8AA5/pdb] (TnsB), 4FCY [https://doi.org/10.2210/pdb4FCY/pdb] (MuA), 3OY9 [https://doi.org/10.2210/pdb3OY9/pdb] (PFV integrase), 6RWL [https://doi.org/10.2210/pdb6RWL/pdb] (SIVrcm integrase). Source data are provided as a Source Data file. Source data are provided with this paper.

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

## Acknowledgements

The authors are grateful to María Cantero Gil for her work during initial IstA characterization, and to the EM platforms of the CIB Margarita Salas, CNB and IBMB (CSIC) for their assistance during grid screening and preparation (especially to Rafael Núñez, Rocío Arranz, Teresa Bueno and Pablo Guerra). We also acknowledge Diamond for access and support of the Cryo-EM facilities at the UK national electron bio-imaging center (eBIC proposal BI26399, to E.A.-P.), funded by the Wellcome Trust, MRC, and BBSRC. Access to eBIC was supported by Instruct-ERIC (PID 10284, to E.A.-P.) and initial data processing was supported by Instruct-ERIC (PID 12164, to E.A.-P.). This work was supported by grant GM071747 (to J.M.B.), grants PID2020-120275GB-I00 and BFU2017-89143-P funded by MCIN/AEI/10.13039/501100011033 and by "ERDF A way of making Europe" (to E.A.-P.), and grant PRE2018-086026 funded by MCIN/AEI/10.13039/501100011033 and by "ESF Investing in your future" (to E.A.-P. and A.G.).

## Author contributions

E.A.-P. and J.M.B. conceived the study. M.S.-A., L.A.-B., A.G. and E.A.-P. performed the integration assays. L.A.-B. and M.S.-A. produced the IstA mutants. L.A.-B. and E.A.-P. characterized the protein-DNA binding properties. M.S.-A. purified the IstA-donor DNA complex and prepared the cryo-EM grids. M.S.-A. collected, processed and analyzed the cryo-EM data. All results were discussed and evaluated with all authors. E.A.-P. supervised the project and wrote the manuscript with input from all the authors.

## Competing interests

The authors declare no competing interests.
