## [Peer Review File · Nature Communications]

IS21 family transposase cleaved donor complex traps two right-handed superhelical crossings.REVIEWER COMMENTS

Reviewer #1 (Remarks to the Author):

This manuscript describes the basic biochemical characterization of an intriguing IS element transposase and its cryoEM structure in complex with IS element ends (its “transpososome”). The similarities and differences when compared to recent structures of Tn7 family transpososomes are interesting from mechanistic and evolutionary viewpoints. The most interesting feature is that the protein – protein interactions in some places reflect those of simpler dimeric transposases such as Mos1, and in other places reflect the more complicated, also tetrameric (in their minimal form) Tn7s and MuA. Additionally, this structure predicts that negative supercoiling should promote and/or stabilize the initial synaptic complex. Overall, this is a nice addition to the transpososome zoo.

Specific comments:

Last line of abstract: sadly, this structure does not actually shed any new light on why this group of transposases rely (to varying degrees) on dedicated ATPase accessory factors for strand transfer. See comment on line 425 / figure 7b below.

Line 74 – Can this transposase really perform simple insertions, or could the observation of simple insertions in in vivo assays be due to homologous recombination resolving cointegrates before detection?

Line 78 – what does it mean to “prime” the target DNA?

Line 115 – please specify the configuration of the ends here – as noted for other transposases, and later in this paper, it matters.

In general, it would help to give an explicit mapping of the oligos in supplementary table 2 to the experiments they were used in (e.g. state something like “substrate ___ for the experiment in Figure ___ was annealed from oligos L-TIR-fwd and …”).

Lines 137-138: Not 100.00% true: for Tn7-like transposases and Mu, the two ends are more different. For example, the right end is preferred for mu and complexes with 2 left ends are disfavored.

Line 157: the Mizuuchi lab showed that this is the preferred configuration for Mu as well.

Paragraph starting with line 174: did these ends have the pre-cleaved configuration? I appreciate that the authors don’t read too much into the Hill coefficient of 3.6, but it may be worth mentioning the difficulties in interpreting it: this is not a simple system with 1 tumbling constant for the unbound and a single slower tumbling constant for the bound species – a 2-subunit-bound DNA will tumble more slowly than a 1-subunit-bound DNA, and the 2-subunit-bound species may be in equilibrium with a 4-protein-2-DNA transpososome, which would give a much higher anisotropy value than the 2-subunit-bound duplex.

Line 187 – precleaved ends?

Line 320 – The DNA axis is deflected from straight by 70 degrees. I see why the authors labeled it as 110 degrees, but by the same logic, one could describe a perfectly straight piece of DNA as being bent by 180 degrees.

Line 324 – 326 or so: I don’t understand the logic, and the HTH2 domains of the upper and lower domains don’t actually appear to be on the same face of the double helix.

Lines 380 – 387 or so: Mu’s left end is more complicated. See Figure 4 of the Montano

paper.

Line 407ish: the promoter formed by IS element tandem repeats is quite interesting. Is it known how the tandem repeats are formed?

Line 425 / figure 7b: this model is the only part of the work that I think needs to be redone. There are now 3 related but target-bound transpososome structures to compare this one with, and they all show grossly distorted DNA between attack sites. Therefore it makes no sense to assume that the protein, not the DNA, must adjust to allow strand transfer in this case. Sadly, I really don't think this new IS21-family structure sheds any new light at all on the need / function for the ATPase protein.

Figure 1a, b: the handedness of the DNA-DNA crossings in the cartoons is incorrect for negative supercoiling.

Figure 4b: (minor artistic issue): the white DNA is drawn in a way that looks very much like the target DNA in Tn7 and Mu structures, which is quite confusing.

Line 506 – what kind of DNA sequencing? A little more detail here would be good.

Minor typos and grammatical issues:

Line 131: plasmids product should be plasmid products

Line 132: ends should be end

Line 134: "the transposase can recognize both ends indistinctly" would be clearer as "the transposase does not distinguish between the transposon ends"

Lines 150 – 153: scientifically quite nice, but very awkwardly described. Needs to be reworded by a native speaker of English.

Line 205 – 5 bases long should be 5 base long (use the singular when making a compound noun)

Line 450 – ion exchange not ionic exchange

Line 510 was not were

Reviewer #2 (Remarks to the Author):

Spinola-Amilibia et al., Manuscript NCOMMS-22-38620

Dr. Arias-Palomo and co-workers present a cryo-EM structure of the IstA DNA transposase of the widespread IS21 family of mobile elements. The result is novel, as this is (to my knowledge) the first such complex solved. The main finding of the work is that the IstA bound to transposon ends shows that the protein assembles into a highly intertwined tetramer that joins two supercoiled terminal inverted repeats. The complex looks very similar to that of the strand-transfer complex of the phage MuA transposase (Montano et al., Nature 2012) and the TnsB transposase from CRISPR-Cas type V-K associated transposons (Park et al., PNAS 2022 and Tenjo-Castano et al., Nature Comms 2022), suggesting that the molecular architecture of the complexes suggesting a mechanistic relationship between these transposition systems.

Unfortunately, the paper as written is somewhat dull, describing a large number of structural details without putting them in a biological and/or mechanistic context. The authors do not venture to address a detailed comparison between IstA and the recent TnsB structures (Kaczmarek et al., Mol. Cell 2022, Park et al., PNAS 2022 and Tenjo-Castano et al., Nature Comms 2022), which could be of interest for the reader and the field, especially (Tenjo-Castano et al Nature Comms 2022) which presents a resolution that allows a detailed comparison. One of the most exciting and outstanding question in the field is how the transposase integrates the cargo DNA. It is well known that this feature is somehow dependent on the dynamic interaction between transposase and the

key ATPase regulator of the system, in this case IstB. I'm surprised that this interaction is not commented in the Discussion, This omission seems especially pointed, given the rapidly emerging structural and mechanistic understanding of the key ATPase regulator in other transposon systems.

Specific points

1.- This is a structural paper, however not a single map of the structure is shown in the figures that allow the reader an assessment of the data quality. The authors show the surface representation of the DNA in the figs. Only fig5b shows the map on the DNA. There is no map of the important protein regions of the structure shown in Fig 4, 5 or 6. How can the reader calibrate the authors conclusions?

2.- P10/L206-209. The selection of only one transposon end for the structural studies in order to suppress heterogeneity and the application of C2 symmetry to improve the maps implies a lack of mechanistic information on the more natural scenario. Furthermore, the fact that IstA uses both ends indistinctly in vitro does not mean that this is the situation in vivo. Therefore, I think the authors should include the maps of the structure with the two ends, even though they may not reach a decent resolution. Sometimes these low-resolution can be quite informative.

3.- P10/L212-215. This statement does not match with the local resolution map in supp fig 3f. Besides the upper part of the map the rest seems to be at the same resolution. Or the local resolution is not properly calculated, or the assertion does not match the map.

3.- No site directed mutagenesis and a subsequent structure-function analysis has been performed in the residues identified as critical for binding or catalysis. This extra data will help to understand and validate the DNA binding regions.

Minor

P9/L203 "perpare"

P11/L229 include TnsB

P12/L270-273, cite Park et al., PNAS 2022 and Tenjo-Castano et al., Nature Comms 2022

P13/L292, cite Tenjo-Castano et al., Nature Comms 2022

Response to Referees

We are grateful to the referees for their time and constructive comments and are gratified by the overall favorable impressions. Additional information, new figures, and changes to improve clarity have been included in the revised text. Answers to specific questions or comments are discussed below in blue text.

Reviewer #1 (Remarks to the Author):

This manuscript describes the basic biochemical characterization of an intriguing IS element transposase and its cryoEM structure in complex with IS element ends (its “transpososome”). The similarities and differences when compared to recent structures of Tn7 family transpososomes are interesting from mechanistic and evolutionary viewpoints. The most interesting feature is that the protein – protein interactions in some places reflect those of simpler dimeric transposases such as Mos1, and in other places reflect the more complicated, also tetrameric (in their minimal form) Tn7s and MuA. Additionally, this structure predicts that negative supercoiling should promote and/or stabilize the initial synaptic complex. Overall, this is a nice addition to the transpososome zoo.

Specific comments:

Last line of abstract: sadly, this structure does not actually shed any new light on why this group of transposases rely (to varying degrees) on dedicated ATPase accessory factors for strand transfer. See comment on line 425 / figure 7b below.

We regret if we somehow overstated this point in the previous version of the manuscript. Although the results presented in this work do not allow to assign an unambiguous role to the IstB ATPase, they do provide insights on why the pre-transposition configuration of IstA described here may be inactive. Modelling the DNA substrates found in the MuA and TnsB strand-transfer complex (STC) structures in IstA shows that the configuration of the IS21 transposase in the pre-transposition state would generate shorter direct repeats than the ones observed experimentally in this work and elsewhere (Xu, et al. Plasmid. 1993) (new Supplementary Fig. 11). This comparison, combined with the observation that the absence of IstB leads to aberrant direct repeats of variable length *in vivo* (Schmid, Journal of Bact, 1999), suggests that the accessory ATPase subunit may serve to help align the target DNA in the active site. Whether this alignment happens by reshaping the target DNA and/or the transposase awaits further experimental analysis. We have down-weighted and clarified our consideration of this issue by editing the Discussion, Abstract, and Figure 7, and by adding a new Supplementary Fig. 11.

Line 74 – Can this transposase really perform simple insertions, or could the observation of simple insertions in *in vivo* assays be due to homologous recombination resolving cointegrates before detection?

We are grateful to the reviewer for raising this question. The precise mechanism of action of this family of transposable elements is not completely clear. Some authors

have suggested that IS21 could employ a non-replicative pathway, whereas others have proposed that it probably uses a copy-out/paste-in mechanism mediated by circular dsDNA intermediates (similar to the one described for the IS3 family (Sekine Y, et al. J. Mol. Biol. 1999)). However, in this particular section of the manuscript, we did not intend to describe a precise mechanism of action but rather indicate that full-length IstA can act on two different substrates (single and tandem IS21 elements) with similar levels of activity. We have edited the text to clarify this point.

Line 78 – what does it mean to “prime” the target DNA?

We apologize for the confusion on this point. Although the molecular details are not yet clear, IstB (similar to other ATPases involved in DNA transposition) helps select/mark target DNAs and recruit the IstA transposase to insertion sites. We have modified this sentence in the revised manuscript.

Line 115 – please specify the configuration of the ends here – as noted for other transposases, and later in this paper, it matters.

All donor DNA substrates used in our biochemical experiments correspond to linear, pre-cleaved, and fully intact (55 bp) TIR sequences unless indicated otherwise. This information has been now included in the text.

In general, it would help to give an explicit mapping of the oligos in supplementary table 2 to the experiments they were used in (e.g. state something like “substrate ___ for the experiment in Figure ___ was annealed from oligos L-TIR-fwd and ...”).

We regret if the manuscript was unclear on this issue. As requested, we have added more detailed information to Supplementary Table I and mapped the oligos to the experiments where they were used.

Lines 137-138: Not 100.00% true: for Tn7-like transposases and Mu, the two ends are more different. For example, the right end is preferred for Mu and complexes with 2 left ends are disfavored.

We apologize for the confusion. We intended to state that some transposases can use the isolated transposon ends *in vitro* in the integration reaction (sometimes with different efficiencies, although the difference is modest in the case of IstA). The strand transfer reaction of Mu is reported to be more efficient with two right ends (especially when pre-cleaved substrates are used) (Craigie and Mizuuchi, 1987; Namgoong et al., 1994), and the overall configuration of left and right ends are indeed substantially different in Tn7 and Mu. We apologize if this part of the manuscript was unclear. The text has been revised to clarify this point.

Line 157: the Mizuuchi lab showed that this is the preferred configuration for Mu as well.

We thank the refer for catching this omission. Although the protein and DNA components of Mu contain additional regulatory elements compared to the IS21 system, the core factors that are capable of performing the strand transfer reaction are remarkably similar. This point was previously mentioned in the Discussion but we

have now referenced it in the Result sections of the new version of the manuscript as well.

Paragraph starting with line 174: did these ends have the pre-cleaved configuration?

The nucleic acid molecules used in the DNA binding experiments did indeed have a pre-cleaved configuration. We tested pre-cleaved DNA molecules with either blunt ends or a 5' overhang, but the fluorescence anisotropy, size exclusion, and negative staining results were identical for the substrates. This finding indicates that although the presence of a small 5' overhang modestly impacts the activity of integration reaction, it does not significantly affect the affinity of IstA for the DNA or the oligomerization of the transposase. These points are now indicated in the text and in Supplementary Table I.

I appreciate that the authors don't read too much into the Hill coefficient of 3.6, but it may be worth mentioning the difficulties in interpreting it: this is not a simple system with 1 tumbling constant for the unbound and a single slower tumbling constant for the bound species – a 2-subunit-bound DNA will tumble more slowly than a 1-subunit-bound DNA, and the 2-subunit-bound species may be in equilibrium with a 4-protein-2-DNA transpososome, which would give a much higher anisotropy value than the 2-subunit-bound duplex.

We thank the reviewer for pointing out this nuance. We agree that it is not straightforward to deconvolute these fluorescence anisotropy data due to the presence of different populations and that the results should not be overinterpreted. We have revised the text to stress this point.

Line 187 – precleaved ends?

The referee is correct in assuming that pre-cleaved DNA substrates were used for the DNA binding and complex assembly experiments. We have revised the text to clarify this point and have added this information in the Supplementary Table I as well.

Line 320 – The DNA axis is deflected from straight by 70 degrees. I see why the authors labeled it as 110 degrees, but by the same logic, one could describe a perfectly straight piece of DNA as being bent by 180 degrees.

We thank the reviewer for bringing this to our attention. The 110° angle was labeled in the figure for convenience and visual clarity. However, we agree that it would be more appropriate to use the supplementary angle. We have incorporated the change in the text and in Fig. 6.

Line 324 – 326 or so: I don't understand the logic, and the HTH2 domains of the upper and lower domains don't actually appear to be on the same face of the double helix.

We apologize if this point was unclear. We intended to highlight a previous suggestion that the spacing between the direct repeats of the IS21 transposon end could favor the formation of supercoiled DNA structures (Berger and Haas. Cell. Mol.

Life Sci. 2001). It has been noted that sequence periodicities which coincide with the helical periodicity of DNA are often associated with global DNA curvature induced by local bending (Trifonov EN, *et al.* PNAS. 1980). We have simplified and edited this part of the manuscript.

Lines 380 – 387 or so: Mu's left end is more complicated. See Figure 4 of the Montano paper.

The DNA regulatory elements found in Mu are indeed more complex than the ones present in IS21. Beside the internal enhancer regions and the presence of specific sequences for the DNA bending proteins HU and IHF, Mu's transposon ends contain three left and three right repeats separated by variable lengths. We apologize if we did not convey this information clearly. This part of the manuscript has now been edited.

Interestingly, although Mu's left end is more complicated, the IstA structure we have captured does match the topological description of a proposed (but not yet imaged) MuA state (Yin *et al.* PNAS. 2005; Harshey *et al.* Crit. Rev. Biochem. Mol. Biol. 2006; Montañó *et al.* Nature. 2012). A configuration for MuA similar to the one described here for IstA would allow for more extensive protein-protein contacts between all the subunits in the tetramer, which might help to explain the extraordinary stability of MuA transpososomes (e.g., resistance to 4 M urea, 6 M NaCl or 65 °C) (Surette *et al.* Cell. 1987). Future studies will be required to elucidate the structure of the MuA oligomers at different stages of the transposition reaction, however.

Line 407ish: the promoter formed by IS element tandem repeats is quite interesting. Is it known how the tandem repeats are formed?

This is an intriguing question indeed. Pre-existing IS21 tandems can integrate close to other IS21 elements *in vivo*, generating new tandems in the target DNA in the process. This observation, however, contravenes the hypothesis that IS21 elements – similarly to Mu or Tn7 – might possess target immunity. However, it is still unclear whether these kinds of tandem mobilizations are mediated by the transposase or whether they might arise from recombination events. How IS21 tandems originate in the first place is still not fully understood. There are different possibilities for their formation, such as when a single IS21 element inserts close to a second one; however, IS21 tandem formation has been only experimentally observed following the spontaneous deletion of a segment situated between two IS21 copies present in direct orientation (Reimann and Haas. Genetics. 1987).

Line 425 / figure 7b: this model is the only part of the work that I think needs to be redone. There are now 3 related but target-bound transpososome structures to compare this one with, and they all show grossly distorted DNA between attack sites. Therefore it makes no sense to assume that the protein, not the DNA, must adjust to allow strand transfer in this case. Sadly, I really don't think this new IS21-family structure sheds any new light at all on the need / function for the ATPase protein.

We agree that it cannot be assumed that only the transposase, and not the target DNA, has to be remodeled to facilitate strand transfer. We regret if the original manuscript emphasized this idea. As requested, we have now compared the

distance between the active sites of related transposases and modelled the STC substrates of the MuA and TnsB structures into our IstA state. Although the experiments reported here do not establish a precise role for the IstB ATPase, the analysis of the structural and biochemical data reported here suggests that isolated IstA is inactive, and that IstB may be involved realigning the target DNA to provide an optimal configuration for the active site. We have edited the text in the Discussion, Abstract, and Figure 7 and have incorporated a new Supplementary Fig. 11 to address this point.

Figure 1a, b: the handedness of the DNA-DNA crossings in the cartoons is incorrect for negative supercoiling.

We thank the reviewer for pointing this out. The figure has been corrected in the new version of the manuscript.

Figure 4b: (minor artistic issue): the white DNA is drawn in a way that looks very much like the target DNA in Tn7 and Mu structures, which is quite confusing.

The figure has been edited to clarify that the nucleic acid corresponds to the donor DNA.

Line 506 – what kind of DNA sequencing? A little more detail here would be good.

We apologize if we did not provide sufficient detail on this matter. The products of the insertion reaction consist of linearized target plasmids that contain one TIR and a direct repeat (generated as a consequence of the strand-transfer reaction) on each end. These products were ethanol precipitated and ligated with the KanR gene. This re-circularization reaction positions the TIRs and direct repeats so as to flank the KanR gene. The plasmids were then transformed into XL1 blue cells and plated into Kan plates. The purified plasmids from ten colonies were subjected to Sanger sequencing using primers that annealed with the ends of the KanR gene, giving an accurate reading of the flanking TIRs and direct repeats. We have edited the Methods section to clarify this issue.

Minor typos and grammatical issues:

Line 131: plasmids product should be plasmid products

The sentence has been fixed in the revised manuscript.

Line 132: ends should be end

This line has been corrected.

Line 134: “the transposase can recognize both ends indistinctly” would be clearer as “the transposase does not distinguish between the transposon ends”

The sentence has been edited as suggested.

Lines 150 – 153: scientifically quite nice, but very awkwardly described. Needs to be reworded by a native speaker of English.

We have revised this part of the text.

Line 205 – 5 bases long should be 5 base long (use the singular when making a compound noun)

We thank the reviewer for pointing out this error. It has been changed in the new version of the manuscript.

Line 450 – ion exchange not ionic exchange

This sentence has been fixed.

Line 510 was not were

The typo has been corrected.

Reviewer #2 (Remarks to the Author):

Spinola-Amilibia et al., Manuscript NCOMMS-22-38620

Dr. Arias-Palomo and co-workers present a cryo-EM structure of the IstA DNA transposase of the widespread IS21 family of mobile elements. The result is novel, as this is (to my knowledge) the first such complex solved. The main finding of the work is that the IstA bound to transposon ends shows that the protein assembles into a highly intertwined tetramer that joins two supercoiled terminal inverted repeats. The complex looks very similar to that of the strand-transfer complex of the phage MuA transposase (Montano et al., Nature 2012) and the TnsB transposase from CRISPR-Cas type V-K associated transposons (Park et al., PNAS 2022 and Tenjo-Castano et al., Nature Comms 2022), suggesting that the molecular architecture of the complexes suggesting a mechanistic relationship between these transposition systems.

Unfortunately, the paper as written is somewhat dull, describing a large number of structural details without putting them in a biological and/or mechanistic context. The authors do not venture to address a detailed comparison between IstA and the recent TnsB structures (Kaczmarek et al., Mol. Cell 2022, Park et al., PNAS 2022 and Tenjo-Castano et al., Nature Comms 2022), which could be of interest for the reader and the field, especially (Tenjo-Castano et al Nature Comms 2022) which presents a resolution that allows a detailed comparison. One of the most exciting and outstanding question in the field is how the transposase integrates the cargo DNA. It is well known that this feature is somehow dependent on the dynamic interaction between transposase and the key ATPase regulator of the system, in this case IstB. I'm surprised that this interaction is not commented in the Discussion, This omission seems especially pointed, given the rapidly emerging structural and mechanistic understanding of the key ATPase regulator in other transposon systems.

We very much agree on the interest of the comparison with the Tn7-related systems. At the time of the manuscript preparation and submission, some of the articles describing the structures of the tetrameric Tn7 transposase had not yet been published and/or the coordinate files were not publicly available. IstA shares most of the domains found in TnsB, the Tn7 transposase. A tetrameric structure recently reported for TnsB with DNA (Park J-U, *et al.* PNAS. 2022; Tenjo-Castaño, *et al.* Nat. Commun. 2022) turns out to be intriguingly similar to the synaptic complex described for IstA in this work. Interestingly, however, the HTH domains of the non-catalytic monomers and the L2/R2 terminal repeats are not ordered in the new TnsB model and thus it is still unclear how the Tn7 transposome engages the multiple terminal repeats present in these mobile elements. We have now revised the manuscript, modified Fig. 7, and added new Supplementary Figs. 10 and 11 to incorporate a more detailed comparison between these transposases.

Very recently, some articles describing the recruitment complex and holo-transposome of Tn7-like elements have been published (Schmitz M, *et al.* Cell. 2022; Park J-U, *et al.* Nature. 2022). Interestingly, these manuscripts report how the regulatory AAA+ factor, TnsC, recognizes the target DNA and couples other effector proteins with the transposase. IstB bears significant functional and structural similarities with the Tn7 ATPase. However, the absence of auxiliary effector proteins in IS21, the unique oligomerization state and three-dimensional structure of IstB, and the shorter C-terminal tail of IstA (as compared to the C-terminal domains of TnsB and MuA that are involved in ATPase interactions) strongly suggest that, although there probably are some similarities, the ATP-dependent recruitment and activation mechanism of IS21 transposases likely possess unique features. Experiments are underway to try to answer this question, but are beyond the scope of the present work. This issue is also discussed in the new version of the manuscript.

Specific points

1.- This is a structural paper, however not a single map of the structure is shown in the figures that allow the reader an assessment of the data quality. The authors show the surface representation of the DNA in the figs. Only fig5b shows the map on the DNA. There is no map of the important protein regions of the structure shown in Fig 4, 5 or 6. How can the reader calibrate the authors conclusions?

We regret that we did not provide sufficiently detailed views of the EM maps in the initial version of the manuscript. Besides the previous Fig. 5b and Supplementary Fig. 3, we have now included close up views of multiple relevant protein and DNA regions of the cryo-EM map in the new Supplementary Fig. 5 to illustrate the quality of the model.

2.- P10/L206-209. The selection of only one transposon end for the structural studies in order to suppress heterogeneity and the application of C2 symmetry to improve the maps implies a lack of mechanistic information on the more natural scenario. Furthermore, the fact that IstA uses both ends indistinctly *in vitro* does not mean that this is the situation *in vivo*. Therefore, I think the authors should include the maps of the structure with the two ends, even though they may not reach a decent resolution. Sometimes these low-resolution can be quite informative.

We very much appreciate this question. Although the sequence, spacing, number of repeats of the IS21 transposon ends are highly similar (Supplementary Fig. 8) compared to other transposable elements, such as Mu or Tn7, we agree that the observation that IstA shows similar levels of activity with both transposon ends *in vitro* does not indicate that they might have the exact same role *in vivo* (and, concordantly, we have not implied this idea in the manuscript).

Nonetheless, using a mixture of left and right ends in the structural studies can generate two potential complications. First, the formation of the paired-end complex using an equimolar mix of left and right ends would generate a heterogeneous combination of left-right (~50 %), left-left (~25 %) and right-right (~25 %) complexes. Second, if the differences between the left and right end halves of transposase are not significant (as suggested by the DNA sequence and negatively stained 2D classes), then image processing can generate spurious pseudo-symmetric artifacts as a consequence of processing in C1 (i.e., no symmetry imposed) which could hamper the three-dimensional classification and the interpretation of different types of structures.

Thus, we decided to use just one of the two transposon ends for the cryo-EM analysis – an approach that has been used in numerous prior structural studies (e.g., Montañó *et al.* Nature 2012; Richardson *et al.* Cell 2009; Ghanim *et al.* NSMB 2019; Park *et al.* PNAS 2022) – to simplify data processing and improve the quality of the results. Importantly, we came to this decision after we already determined that IstA's transposition activity, DNA binding capacity, and ability to form paired-end complexes (as judged by SEC and negative-stain EM) was comparable regardless of whether an isolated left TIR, an isolated right TIR, or a mixture of both TIRs was employed. The cryo-EM reconstruction further supports this observation by showing that the transposase primarily establishes specific contacts with regions that are conserved between the left and right ends (termed CS1 and CS2 in this work), whereas the less conserved regions mainly make non-specific interactions or are solvent exposed.

To further test the equivalency of the two TIR ends in supporting IstA function, we have now collected a cryo-EM dataset of IstA in complex with an equimolar mixture of left and right TIRs. Importantly, although the resolution and quality of the reconstruction is lower (due at least in part to the presence of a higher degree of compositional heterogeneity), the 2D class averages and three-dimensional reconstruction establish that the overall configuration of IstA paired-donor complexes formed with either the isolated right TIR end or a mixture of left and right TIR ends is remarkably similar. This new experiment and information are now discussed in the text and in a new Supplementary Fig. 3.

3.- P10/L212-215. This statement does not match with the local resolution map in supp fig 3f. Besides the upper part of the map the rest seems to be at the same resolution. Or the local resolution is not properly calculated, or the assertion does not match the map.

We apologize for the confusion. We have now edited the panel and rendered the map using the same threshold used in Supplementary Fig. 4d to allow for direct comparison.

3.- No site directed mutagenesis and a subsequent structure-function analysis has been performed in the residues identified as critical for binding or catalysis. This extra data will help to understand and validate the DNA binding regions.

We agree with the referee and have cloned, purified, and tested the effects of several IstA mutants that we predict (based on the structure) to be involved in DNA recognition and catalysis. Specific mutations include residues implicated in the recognition of the CS1 and CS2 boxes (R31A, K32A, T92A and K95A), which we now find to have a significant reduction in integration activity. This result highlights the relevance of the contacts between the HTH-1 and HTH-2 domains with the donor DNA. In addition, we have mutated the catalytic triad in IstA (D124A, D187A and E233A) that is responsible for performing DNA breakage/reunion reactions; these mutants all severely compromise the observed integration activity of IstA. Finally, we mutated residues that stabilize the non-transferred strand (i.e., the 5' overhang) in the structure. Although we were unable to produce a Y113A mutant despite several attempts (possibly due to peculiarities of the nucleotide sequence in this region), a Q369A variant did show an appreciable negative effect on IstA activity. We discuss these new experiments in the text and in the new Fig. 5e and Fig. 6d.

Minor
P9/L203 “perpare”

We have edited the text to correct this error.

P11L229 include TnsB

We have included TnsB and the corresponding references in the description of the DNA binding domains.

P12/L270-273, cite Park et al., PNAS 2022 and Tenjo-Castano et al., Nature Comms 2022

The references have been now incorporated in the text.

P13/L292, cite Tenjo-Castano et al., Nature Comms 2022

This recent publication has also been added to the manuscript.

REVIEWERS' COMMENTS

Reviewer #1 (Remarks to the Author):

In this revised version the authors have satisfied all of my previous questions and concerns.

Reviewer #2 (Remarks to the Author):

I read the new version of the paper and the reply to my questions. the authors have done a good job and, in my opinion, I think the paper is ready for publication.

I've no further queries

Response to Referees

Answers to specific questions or comments are discussed below in blue text.

Reviewer #1 (Remarks to the Author):

In this revised version the authors have satisfied all of my previous questions and concerns.

We are grateful to the reviewer for the current and previous comments.

Reviewer #2 (Remarks to the Author):

I read the new version of the paper and the reply to my questions. the authors have done a good job and, in my opinion, I think the paper is ready for publication.

I've no further queries.

We thank the reviewer for the kind words and for the previous comments and questions. The suggestions made by both referees definitely had a positive impact on the manuscript.